# Predicting the Number of People In a Room

**Kuanghao Wang**
2024310033
School of Architecture
Tsinghua University
Haidian District, Beijing
`wkh24@mails.tsinghua.edu.cn`

## Abstract

At present, the energy consumption of the whole building process and the energy consumption of building operation in China accounts for a high proportion of the country's total energy consumption, and reducing the energy consumption of building operation is an important part of saving energy, reducing greenhouse gas emissions, mitigating climate change and achieving carbon neutrality. Building energy consumption is mainly composed of two parts: cooling system energy consumption and heating system energy consumption, both of which are related to indoor cooling and heating loads. A large part of the indoor load is generated by indoor personnel, and the load has a lagging effect. Therefore, if the number of indoor personnel can be predicted in advance, the start/stop time and operating power of the cooling and heating systems can be determined in advance, so as to avoid wastage of cold and heat, and thus to achieve the purpose of energy saving and emission reduction. In the past, there are fewer studies on the time series prediction of indoor occupancy, so this paper will take a specific room as the object of study, based on its occupancy monitoring data, use different methods such as Support Vector Machine (SVM), Auto Regressive Integrated Moving Average (ARIMA), Random Forest, Deep Neural Networks, etc. to predict the number of indoor occupants, and evaluate the advantages and disadvantages of the effectiveness of various prediction methods.

## 1 Introduction

### 1.1 Research background

Energy consumption in building operation mainly includes energy consumption for heating, cooling, ventilation, hot water, lighting and other household appliances during the use of the building, etc. In recent years, with the rapid development of the economy and the continuous improvement of the people's requirements for the indoor environment, the energy consumption in building operation has been increasing year by year, and it has now accounted for about 21.9% of the country's total energy consumption [1]. In developed countries such as in Europe, the air-conditioning system related to the creation of indoor environment has reached 65% of energy consumption in building operation, accounting for a relatively large proportion. In order to save energy and reduce emissions, and to achieve the goals of carbon peaking and carbon neutrality, it is particularly critical to reduce energy consumption in building operations, especially air-conditioning energy consumption.

In summer, for example, the indoor heat load mainly comes from heat transfer from the enclosure structure, heat generation from indoor equipment and heat generation from indoor personnel. Due to the hysteresis of the heat load, when the number of indoor personnel changes, the indoor heat load and the air conditioning cooling capacity will not be completely matched, which leads to the inability to

38th Conference on Neural Information Processing Systems (NeurIPS 2024).

accurately control the indoor temperature, so that the indoor personnel's thermal comfort evaluation is reduced. On the other hand, there is blindness in adjusting the operating power of the cooling system according to the number of existing people. If the number of indoor personnel continues to increase, the cooling power will always be insufficient, resulting in increasing indoor temperature and poor cooling effect; if the number of indoor personnel remains unchanged or decreases, the excess cooling capacity will result in a lower indoor temperature, increasing the operational energy consumption. Existing control methods in order to make the indoor personnel thermal comfort good, mostly used as much as possible to enhance the cooling capacity to cope with changes in the number of indoor personnel. In this case, when the cooling capacity is high, more heat can be introduced by the indoor people themselves by opening windows and ventilating the room, etc. The mixing of heat and cold caused here also reflects the waste of energy. Therefore, if the number of people in the room can be predicted in the short term, so that the air conditioning operation strategy can be formulated in advance to cope with the excess heat load in advance, not only can it provide a more stable and comfortable thermal environment for the people in the room, but also save a lot of energy to avoid the excess heat and cold mixing at a later stage.

## 1.2 Domestic and foreign research status

For this content, domestic and foreign scholars have already had some research results.Dong [2] et al. deployed an Information Technology Enabled Sustainability Testbed (ITEST) in a smart open-plan office building. Indoor environmental parameters were measured using eight different sensors, and the number of people in the room was predicted using the Hidden Markov Model (HMM) method, with the accuracy of the predicted results lying between 65% and 80%.Candanedo and Feldheim [3] used a multi-person office as an experimental site for the collection of indoor environmental parameters such as CO2 concentration, temperature and humidity and illumination. They used the new model Classification and Regression Tree (CART) proposed by Breiman et al [4] for prediction and obtained results with 96% accuracy.Yang et al [5] used the SVM model for indoor occupancy prediction using six offices in two office buildings on the main campus of the University of Southern California as the experimental sites and achieved a prediction accuracy of 89.6%.

However, it is easy to see that more of these elements are focused on obtaining an accurate dataset of the number of people, and relatively little on further analysing and processing the dataset. At the same time, these datasets are not large enough overall to be measured for very long periods of time. Therefore, this paper will focus on analysing one of the available quality datasets using different methods.

## 1.3 Research methodology and content

Based on the existing research and combining the content learnt in the course, this paper will mainly try to predict the number of indoor occupants using methods such as SVM, ARIMA, Random Forest, Deep Neural Networks, etc. and evaluate the prediction effect of different methods. The dataset of this paper originates from the measurements in the office 201 of Dest Building, School of Architecture, Tsinghua University, which started from 1 April 2022 to 31 March 2024. The measurement method mainly uses image recognition technology. By using the binocular vertical image recognition human displacement sensor, the number of people in the room can be counted by recognising their heads and shoulders. In addition to this, manual adjustments were made every day based on the measured data and the actual data to minimise measurement errors. The data is recorded every 15 minutes, which makes the data volume relatively large. Specifically, in this paper, the existing dataset will be divided into two parts, one part is the training set and the other part is the test set. The test set has only one week's worth of data, which is used as real data for evaluating the predictions.

A preliminary analysis of the available data set shows that the overall number of people varies from Monday to Sunday in a cyclical manner. At the same time there is a pattern in the number of people at the same time in each weekday or weekend, which means that short-term prediction using machine learning is possible. However, on the other hand, this dataset contains two periods of time, during and after the COVID-19, and the number of office personnel is highly random, coupled with the fact that random events such as personnel visits often occur, and the number of people fluctuates over a wide range, making it difficult to make accurate predictions. For the purposes of this paper, a particular method is considered valid as long as the results predicted by this methods is better than the results averaged directly over the same time period.

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
