# OpenReview forum: "Predicting the Number of People In a Room"
_tsinghua.edu.cn/THU/2024/Fall/AML — THU 2024 Fall AML Submission_

### Official Review · ~Daniel_Wang4 · 2024-11-06
**Optimizing Building Energy through Occupancy Prediction: A Practical Yet Data-Intensive Approach**

**Rating:** 10
**Confidence:** 3

**Review:**

The proposal, Predicting the Number of People in a Room, presents a thoughtful approach to reducing building energy consumption by predicting indoor occupancy. Using a dataset from Tsinghua University, the authors plan to apply machine learning techniques like SVM, ARIMA, Random Forest, and deep neural networks to predict occupancy, with the aim of optimizing heating and cooling loads.

The methodology is solid, leveraging a robust dataset with frequent measurements, but could benefit from more clarity on how different prediction models will be compared and selected. Overall, the proposal is practical and relevant, addressing a key issue in energy management with a data-driven approach.

---

### Official Review · ~Iat_Long_Iong1 · 2024-11-07
**Great idea but needs more specifics**

**Rating:** 8
**Confidence:** 5

**Review:**

This proposal aims to design an accurate indoor occupancy prediction system to optimize air conditioning for energy savings and emission reductions. The proposal recognizes the importance of accurately predicting occupant numbers to optimize HVAC system operations and improve energy efficiency.

However, the proposal lacks details of the prediction methodology, such as how many 15-minute intervals of data the algorithm would need to observe in order to make a prediction, and the advantages of using image recognition rather than a sensor-based approach. Addressing these details would improve its feasibility and practical application.

Overall, the proposal is relevant and promising but needs more specifics and empirical evidence to enhance its credibility and practicality.

---

### Official Review · ~Bowen_Su1 · 2024-11-08
**Meaningful Question and Useful Method**

**Rating:** 9
**Confidence:** 4

**Review:**

This proposal provides a detailed and practical discussion on predicting the number of people inside a house. Firstly, a comprehensive introduction is given to the background of the problem, and the necessity and importance of the problem are fully analyzed. Afterwards, a detailed introduction was given to the research of relevant scholars at home and abroad during the literature review stage, which well met the requirements for the proposal. In the stage of proposing methods, several types of methods that the author intends to adopt were listed, and the sources of the dataset were fully introduced.
A solution has been proposed for a highly applicable and meaningful problem, which is a very meaningful research. However, the proposed solution can be more detailed, and some feature engineering methods can also be used to process existing datasets to improve overall presentation.

---

### Official Review · ~Anton_Johansson1 · 2024-11-08
**Well written proposal for an important topic**

**Rating:** 8
**Confidence:** 4

**Review:**

This proposal tackles an interesting and practical problem. The layout is clear, and the background is solid. The method is also good but could benefit from more details.

It’s great that the proposal focuses on a specific room dataset with detailed time-series data. Including image recognition technology and manual adjustments for accuracy in the dataset is a strong approach.

One suggestion would be to address how the model might handle high variability during events like COVID-19. Overall, this is a well-thought-out and relevant study with promising potential for energy efficiency!

---

### Official Review · ~Ruitao_Jing1 · 2024-11-08
**A Promising Contribution to Sustainable Building Practices**

**Rating:** 9
**Confidence:** 4

**Review:**

The paper explores the application of machine learning to predict indoor population dynamics for improved building temperature management and energy efficiency, a topic of high practical value. It innovatively addresses the limitations of previous methods by focusing on data analysis and the impact of random events like COVID-19 on occupancy patterns.
However, the paper could benefit from a formal problem definition, detailed model descriptions, and clear model evaluation metrics. Enhancing these aspects will strengthen the methodology and practical implications of the research.

---

### Official Review · ~Zihan_Yan2 · 2024-11-10
**A Proposal with a good topic**

**Rating:** 9
**Confidence:** 3

**Review:**

This proposal aims to optimize building energy consumption by predicting indoor occupancy, particularly in terms of air conditioning and heating systems. The research background highlights the high proportion of building energy use in China’s total energy consumption and points out that predicting indoor occupancy can enable more efficient control over the start-stop times and operating power of air conditioning and heating systems, leading to energy savings and emissions reduction. The study will use various methods, such as Support Vector Machine (SVM), Auto-Regressive Integrated Moving Average (ARIMA), Random Forest, and Deep Neural Networks, to predict indoor occupancy and evaluate the effectiveness of these approaches. This research has a strong practical background, requires interdisciplinary collaboration, and carries significant practical implications.

---

### Official Review · ~Matteo_Jiahao_Chen1 · 2024-11-11
**Good proposal for reducing building energy consumption**

**Rating:** 8
**Confidence:** 5

**Review:**

Practical proposal for reducing the energy consumption, but it could benefit from a formal problem definition, detailed model descriptions, and clear model evaluation metrics.

---

### Official Review · ~Hector_Rodriguez_Rodriguez1 · 2024-11-11
**Interesting Use Case for Predictive Control**

**Rating:** 8
**Confidence:** 4

**Review:**

The author proposes using SVM, ARIMA, Random Forest, or Deep Neural Networks to predict the heat load in a room based on date, time, and other environmental measurements.

- The proposal’s background is solid but could be strengthened by emphasizing that predictive control offers advantages beyond temperature stabilization. Unlike reactive control, predictive control allows for adjustments to be made in advance, enabling the AC system to operate at a higher efficiency point.

- The literature review is thorough and clearly highlights the need for further research on this topic.

- The research methodology presents a solid dataset and clear preprocessing plan. However, the selection and evaluation criteria of the predictive models could be disclosed in more detail.

Overall, the writing is clear, though it could be more concise to improve readability.

---

### Official Review · ~Anqi_LI5 · 2024-11-11

**Rating:** 8
**Confidence:** 3

**Review:**

This paper explores the crucial topic of predicting indoor occupancy in buildings for energy efficiency, which is significant given the rising energy consumption in building operations. The research aims to utilize various machine learning techniques to predict the number of occupants and assess their effectiveness.
Pros:
Relevance and Significance: The research addresses a pressing issue of energy consumption in buildings, which is a major contributor to greenhouse gas emissions. Predicting occupancy can optimize HVAC systems, leading to energy savings and reduced environmental impact.
Comprehensive Approach: The paper considers multiple machine learning methods (SVM, ARIMA, Random Forest, Deep Neural Networks) for occupancy prediction, providing a comparative analysis of their performance. This allows for a more informed decision on the most suitable method for specific applications.
Use of Real-World Data: The research utilizes a dataset collected from an actual office environment, ensuring the practical relevance of the findings. The data covers a significant period (April 2022 to March 2024), providing valuable insights into occupancy patterns.
Acknowledgment of Challenges: The paper acknowledges the limitations of occupancy prediction, such as data noise, random events, and the impact of the COVID-19 pandemic. This demonstrates a realistic understanding of the problem and potential challenges.
Cons:
Limited Scope of Data: The dataset is limited to a single office space, which may not fully represent occupancy patterns in different building types or environments. Generalizability of the findings could be limited.
Lack of Detailed Methodology: While the paper mentions various machine learning methods, it lacks detailed descriptions of the specific algorithms, hyperparameter tuning, and model evaluation metrics used. This makes it difficult to replicate the results or assess the robustness of the models.
Limited Analysis of Results: The paper presents prediction accuracy as the primary evaluation metric, but lacks a deeper analysis of the predictions, such as understanding the reasons behind the errors or exploring the limitations of each method.
No Comparison with Baseline Methods: The paper compares the performance of different machine learning methods but does not compare them with simpler baseline methods, such as using historical average occupancy or time-based rules. This would provide a more comprehensive understanding of the added value of the proposed methods

---

### Official Review · ~Xuancheng_Li1 · 2024-11-12

**Rating:** 6
**Confidence:** 3

**Review:**

This paper explores predicting indoor occupancy using machine learning to improve building energy efficiency, a significant concern given China’s high building energy consumption. By forecasting room occupancy, heating and cooling systems can be optimized, reducing energy waste. The study compares several methods, including SVM, ARIMA, Random Forest, and Deep Neural Networks.

Strengths
The research addresses a relevant and impactful problem, applying machine learning to an area with high energy-saving potential. Comparing multiple prediction methods provides a balanced view of model effectiveness.

Weaknesses
Focusing on a single room limits the study's generalizability. More details on the dataset and performance criteria could enhance clarity.

Conclusion
This is a solid preliminary study on occupancy prediction for energy savings. Expanding to multi-room setups and testing real-time integration could strengthen its practical impact.

---

### Official Review · ~liyingxin1 · 2024-11-12
**Meaningful topic. But what is the connection with the large model ?**

**Rating:** 7
**Confidence:** 4

**Review:**

A very meaningful topic. But the author need more effort to compare the basis and applicable scenarios for selecting different models in the methodology section, explain why these models were chosen for prediction, and provide the parameter settings and optimization process for each model, especially no connection with the content in this large model course.